# Self-Knowledge Guided Retrieval Augmentation
# for Large Language Models

**Yile Wang[1], Peng Li[*1,3], Maosong Sun[2], Yang Liu[* 1,2,3]**
[1]Institute for AI Industry Research (AIR), Tsinghua University, Beijing, China
[2]Dept. of Comp. Sci. & Tech., Institute for AI, Tsinghua University, Beijing, China
[3]Shanghai Artificial Intelligence Laboratory, Shanghai, China
{wangyile,lipeng}@air.tsinghua.edu.cn, {sms,liuyang2011}@tsinghua.edu.cn

## Abstract

Large language models (LLMs) have shown superior performance without task-specific fine-tuning. Despite the success, the knowledge stored in the parameters of LLMs could still be incomplete and difficult to update due to the computational costs. As complementary, retrieval-based methods can offer nonparametric world knowledge and improve the performance on tasks such as question answering. However, we find that the retrieved knowledge does not always help and even has a negative impact on original responses occasionally. To better make use of both internal knowledge and external world knowledge, we investigate eliciting the model's ability to recognize what they know and do not know (which is also called "self-knowledge") and propose Self-Knowledge guided Retrieval augmentation (SKR), a simple yet effective method which can let LLMs refer to the questions they have previously encountered and adaptively call for external resources when dealing with new questions. We evaluate SKR on multiple datasets and demonstrate that it outperforms chain-of-thought based and fully retrieval-based methods by using either InstructGPT or ChatGPT. Code is released at https://github.com/THUNLP-MT/SKR.

## 1 Introduction

Large language models (LLMs, Brown et al., 2020; Chowdhery et al., 2022; Ouyang et al., 2022) have achieved remarkable performance without much task-specific fine-tuning. However, the full-parametric knowledge stored in LLMs could still be incomplete and difficult to update due to the computational costs. Alternatively, retrieval-augmented methods (Guu et al., 2020; Lewis et al., 2020b; Borgeaud et al., 2022; Izacard et al., 2022; Shi et al., 2023) can utilize external resources such as Wikipedia and offer complementary non-parametric knowledge to enrich the contextualized

---
[*] Corresponding authors.

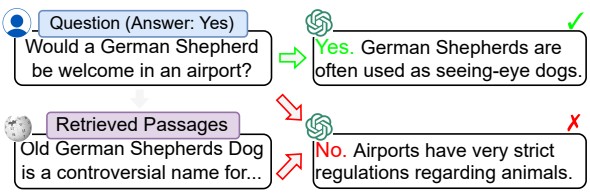

Figure 1: Comparison between two responses given by InstructGPT. The retrieved passages are relevant but not particularly helpful for solving the question, which influences the model's judgment and leads to incorrect answers.

information, thus helping the model generate more reliable answers.

Retrieval augmentation has shown to be very effective for models such as BERT (Devlin et al., 2019), BART (Lewis et al., 2020a), and T5 (Raffel et al., 2020) in various tasks (Karpukhin et al., 2020; Khandelwal et al., 2020, 2021; Izacard and Grave, 2021; Wang et al., 2022; Guo et al., 2023). As LLMs become more and more "knowledgable", recent studies show that the benefit brought from retrieval augmentation is reducing (Mallen et al., 2022; Yoran et al., 2023). Moreover, we find that the retrieved passages could even negatively affect what LLMs originally know. As illustrated in Figure 1, the model can directly give reasonable answers "*German Shepherds are often used as seeing-eye dogs*", however, it is distracted and gives incorrect ones by adding retrieved passages.

The above findings show that one should be more careful when applying the retrieval-based method since it is difficult to know in advance whether the retrieved results are better than what LLMs already captured. To this end, a key issue is to figure out what LLMs do well (e.g., they can answer correctly without assistance) and what they cannot do well (e.g., they answer incorrectly and external information can lead to improved results).

Unfortunately, LLMs themselves have a limited ability to recognize what they know and do not

know, which is also called "self-knowledge" (Yin et al., 2023). However, such an ability is crucial for generating truthful responses (Kadavath et al., 2022) and could be helpful for LLMs themselves to "decide when and when not to use tools" such as a retriever (Mialon et al., 2023).

In this paper, we investigate eliciting the self-knowledge of LLMs and propose a simple yet effective Self-Knowledge guided Retrieval augmentation (SKR) method to flexibly call the retriever for making better use of both internal and external knowledge. In particular, different from existing studies that evaluate the ability through specifically designed metrics or datasets, we collect the self-knowledge of training questions by comparing the performance with or without retrieval augmentation. Then, we propose several strategies to detect the self-knowledge corresponding to a question by referring to the existing collected training questions, including using the LLMs themselves through prompting or explicitly training a small model. Finally, we leverage such elicited self-knowledge to better solve the question through adaptive retrieval augmentation.

We evaluate SKR on five datasets by using InstructGPT (`text-davinci-003`) and ChatGPT (`gpt-3.5-turbo-0301`). Experimental results show that SKR outperforms chain-of-thought based (Wei et al., 2022) and fully retrieval-based methods by 4.08%/2.91% (for InstructGPT) and 4.02%/4.20% (for ChatGPT), respectively.

## 2 Related Work

**Retrieval-Augmented LLMs** Recent studies show that retrieval-augmented methods can enhance the reasoning ability of LLMs (Trivedi et al., 2022; He et al., 2022; Yu et al., 2023; Shao et al., 2023; Jiang et al., 2023) and make the responses more credible and traceable (Xu et al., 2023b; Qian et al., 2023). For example, Trivedi et al. (2022) uses the chain-of-thought (Wei et al., 2022) reasoning steps as queries and uses the results to guide further reasoning and retrieval. He et al. (2022) uses an external natural language inference model to select the most supported reasoning path via retrieved evidence. Yu et al. (2023) propose using the retrieval feedback to refine the output of LLMs to be more reliable and accurate. Xu et al. (2023b) propose search-in-chain and make LLMs interact with retrievers to improve accuracy and credibility. These methods aim at integrating sufficient external knowledge for

a better reasoning process, while we propose to better utilize both the internal and external knowledge through eliciting the self-knowledge of LLMs.

Another line of work tries to teach LLMs to use external tools including retriever, calculator, other foundation models, etc. (Schick et al., 2023; Shen et al., 2023; Qin et al., 2023). These works focus more on leveraging the language understanding capabilities of LLMs to deploy suitable tools in different scenarios, while our work investigates the self-knowledge of LLMs and tries to integrate them with retrievers in a more flexible manner.

**Self-Knowledge in LLMs** "Self-knowledge" in LLMs is originally mentioned in Kadavath et al. (2022), which is used to measure the LLMs' confidence in their own knowledge and reasoning. Such ability is further defined as "the ability to understand limitations on the unknowns" and evaluated by Yin et al. (2023), where they find a considerable gap exists between self-knowledge in models and humans. To explore the LLMs capabilities more extensively, unanswerable and more challenging datasets are also proposed (Rajpurkar et al., 2018; Srivastava et al., 2022; Suzgun et al., 2022). Our work is also related to detecting what LLMs know and do not know, while we do not design new evaluation metrics or challenging datasets to test the ability. By explicitly introducing the external resources, we detect the knowledge boundary of LLMs through the performance changes. Moreover, instead of evaluating each question independently, we propose several ways to elicit self-knowledge by referring to existing cases.

## 3 Method

Our method is depicted under the question-answering settings, which has been a popular way to interact with and assess LLMs. The overall pipeline is shown in Figure 2, which includes collecting, eliciting, and using self-knowledge of LLMs. We introduce each of them as follows.

### 3.1 Collecting Self-Knowledge of LLMs from Training Samples

Given a dataset $\mathcal{D}$ with training question-answer pairs $\{q_j, a_j\}_{j=1}^{|\mathcal{D}|}$, we can use the LLM $\mathcal{M}$ to generate the answers for each question $q_i$ via few-shot in-context learning (Brown et al., 2020):

$$\hat{a}(\mathcal{M}, q_i) = \mathcal{M}(q_1 \circ a_1, ..., q_d \circ a_d, q_i), \quad (1)$$

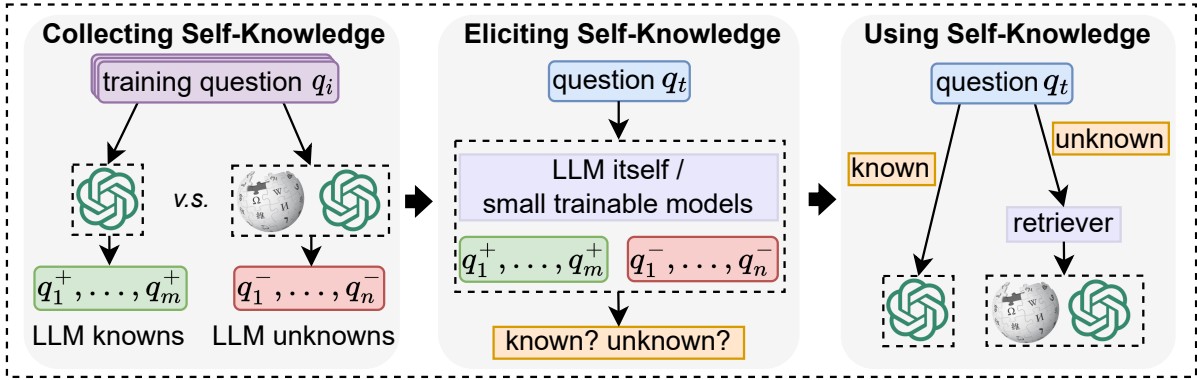

Figure 2: The overall pipeline of our SKR method. We first collect self-knowledge from training questions according to the performance with or without external information (§ 3.1). Then we use the LLMs themselves or explicit small trainable models to elicit self-knowledge of a question $q_t$ by referring to the collected self-knowledge from training questions (§ 3.2). Finally, we use the self-knowledge to the new question and adaptively call a retriever (§ 3.3).

where ○ denotes concatenation and $\{q_j \circ a_j\}_{j=1}^d$ are $d$ demonstrations.

The above generated answers $\hat{a}(\mathcal{M}, q_i)$ reflects the internal knowledge to question $q_i$ in $\mathcal{M}$. Meanwhile, we can possibly find passages from external resources that may be related to $q_i$, such passages can be used as additional information for the model input. Formally, for each question, we first use a pre-trained retriever $\mathcal{R}$ to find the possibly related information from corpus $\mathcal{C}$:

$$p_i = \{p_{i1}, p_{i2}, ..., p_{ik}\} = \mathcal{R}(q_i, \mathcal{C}), \quad (2)$$

where $p_i = \{p_{i1}, p_{i2}, ..., p_{ik}\}$ are the top-$k$ retrieved passages for $q_i$. In practice, we set $\mathcal{R}$ as dense passage retriever (Karpukhin et al., 2020) and $\mathcal{C}$ as passage chunks from Wikipedia. Then, we use $\mathcal{M}$ again to generate the answer with retrieval augmentation:

$$\hat{a}^{\mathcal{R}}(\mathcal{M}, q_i) = \mathcal{M}(q_1 \circ p_1 \circ a_1, ..., q_d \circ p_d \circ a_d, q_i \circ p_i). \quad (3)$$

Given the answers $\hat{a}(\mathcal{M}, q_i)$, $\hat{a}^{\mathcal{R}}(\mathcal{M}, q_i)$, and the ground-truth answer $a_i$, we categorize each question into positive subset $\mathcal{D}^+$ and negative subset $\mathcal{D}^-$ based on the differences between results:

$$q_i \in \begin{cases} \mathcal{D}^+, & \text{if } \mathrm{E}[\hat{a}(\mathcal{M}, q_i)] \geq \mathrm{E}[\hat{a}^{\mathcal{R}}(\mathcal{M}, q_i)]; \\ \mathcal{D}^-, & \text{otherwise,} \end{cases} \quad (4)$$

where E is an evaluation metric such as accuracy and exact match score, we discard the question $q_i$ if both the $\hat{a}(\mathcal{M}, q_i)$ and $\hat{a}^{\mathcal{R}}(\mathcal{M}, q_i)$ are incorrect.

Finally, the training set can be split into subset $\mathcal{D}^+ = \{q_1^+, ..., q_m^+\}$ which includes questions that $\mathcal{M}$ can directly give correct answers without external information (LLM knowns) and the subset

$\mathcal{D}^- = \{q_1^-, ..., q_n^-\}$ where the external information can lead to more accurate results (LLM unknowns).

## 3.2 Eliciting Self-Knowledge of LLMs

Four different strategies are proposed to detect the self-knowledge of target questions, including direct prompting, in-context learning, training a classifier, and nearest neighbor search. We use the LLM itself in the former two methods and explicit smaller modes in the latter two methods.

**Direct Prompting** Given a question $q_t$, a straightforward way to detect whether LLMs are capable of solving it is to ask them directly:

---

**Direct Prompting**

(prompt)
$\{q_t\}$ Q: *Do you need additional information to answer this question?* A:

(possible response)
No, I don't need additional information to answer this question. / Yes, I need additional information to answer this question.

---

Here we use the prompt "*Do you need additional information to answer this question?*" as a template and detect self-knowledge according to the possible response. We thought LLM is capable (or not capable) of solving the question well when they "don't need (or need) additional information". Direct prompting may intuitively work, but it tests each question independently and does not make use of the collected training questions in Section 3.1. To remedy this issue, we further leverage the collected self-knowledge from training questions in the next three strategies.

**In-Context Learning** LLMs have shown a strong capability to learn from demonstrations and infer

through few-shot in-context learning (Brown et al., 2020). We select few training questions from both $\mathcal{D}^+$ and $\mathcal{D}^-$ as demonstrations to elicit the self-knowledge to the question $q_t$:

---

**In-Context Learning**

(prompt)
$\{q_1^+\}$ Q: *Do you need additional information to answer this question?* A: *No, I don't need additional information to answer this question.*
$\{q_1^-\}$ Q: *Do you need additional information to answer this question?* A: *Yes, I need additional information to answer this question.*
......
$\{q_t\}$ Q: *Do you need additional information to answer this question?* A:

(possible response)
No, I don't need additional information to answer this question. / Yes, I need additional information to answer this question.

---

Here we use the answer templates "*No, I don't need...*" or "*Yes, I need...*" in demonstrations based on whether the corresponding question comes from positive set $\mathcal{D}^+$ or negative set $\mathcal{D}^-$, respectively.

The proposed direct prompting and in-context learning methods can elicit self-knowledge of LLMs to some extent. However, they have several limitations. First, both methods require designing prompts and calling the LLMs for each new question, which makes it impractical. Second, in-context learning could also be unstable due to contextual bias and sensitivity (Zhao et al., 2021; Lu et al., 2022) and it is more difficult to address such an issue for close-source LLMs. Third, they cannot make use of all questions due to the constraints of maximum tokens. To make our method more practical and avoid the above issues, we further leverage smaller models to help elicit self-knowledge.

**Training a Classifier** Given $\mathcal{D}^+$ and $\mathcal{D}^-$, we can take them as a two-way classification problem (e.g., setting $q_i$ in $\mathcal{D}^+$ with a positive label and $q_i$ in $\mathcal{D}^-$ with a negative label) and use all the samples to train a classifier such as BERT-base (Devlin et al., 2019) explicitly:

$$\hat{y}_i = \text{softmax}(W h_{\text{cls}}(q_i) + b), \quad (5)$$

where $q_i \in \mathcal{D}^+ \cup \mathcal{D}^-$ is a training question, $h_{\text{cls}}(q_i)$ is the sentence-level representation from BERT-base, $W$ and $b$ are parameters of the classification head. The parameters can be optimized by minimizing the cross-entropy loss between the predicted label distribution $\hat{y}_i$ and the ground-truth label of $q_i$. Once the training is complete, we can infer the label of question $q_t$ similar to Eq. 5.

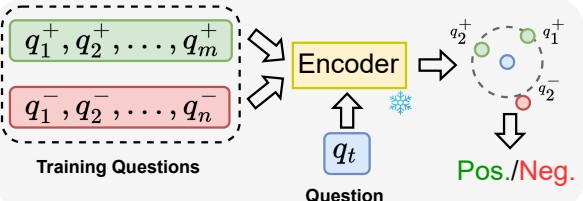

Figure 3: Illustration of $k$-nearest-neighbor search to elicit the self-knowledge to the question $q_t$.

**Nearest Neighbor Search** Instead of training an explicit smaller model, we can infer the label of questions through $k$-nearest-neighbor ($k$NN) search by using a pre-trained fixed encoder, as shown in Figure 3. $k$NN (Fix and Hodges, 1989) is a widely used algorithm and benefit for a range of NLP tasks (Khandelwal et al., 2020, 2021; Shi et al., 2022; Xu et al., 2023a). Our motivation is similar in that if two questions are close in the semantically embedded space, then the LLMs would show similar self-knowledge for both of them.

Formally, we encode each question into embeddings and compute the semantic similarity through cosine distance $\text{sim}(q_t, q_i) = \frac{e(q_t) \cdot e(q_i)}{||e(q_t)|| \cdot ||e(q_i)||}$, where $q_i \in \{q_1^+, ..., q_m^+, q_1^-, ..., q_n^-\}$, $e(\cdot)$ is the representations of a sentence encoder such as SimCSE (Gao et al., 2021). Then we search the top-$k$ nearest neighbors with the highest similarity. If the top-$k$ nearest neighbors include $\ell$ positive ones and $k - \ell$ negative ones, we label the question $q_t$ as positive if $\frac{\ell}{k-\ell} \geq \frac{m}{n}$ or negative if $\frac{\ell}{k-\ell} < \frac{m}{n}$ ($m$ and $n$ are the numbers of questions in $\mathcal{D}^+$ and $\mathcal{D}^-$, respectively).

### 3.3 Using Self-Knowledge for Adaptive Retrieval Augmentation

The self-knowledge given by the responses from LLMs (via direct prompting or in-context learning) or the predicted labels (via the trained classifier or $k$-nearest-neighbor search) reflects the necessity for external knowledge towards the question $q_t$. Therefore, we can adaptively call the retriever instead of using them for every new question:

---

**Adaptive Retrieval Augmentation**

(for LLM known)
$\{q_1 \circ a_1\}, ..., \{q_d \circ a_d\}, \{q_t\}$
A: (LLM directly answers without retrieval)

(for LLM unknown)
$\{q_1 \circ p_1 \circ a_1\}, ..., \{q_d \circ p_d \circ a_d\}, \{q_t\}$
*Here are some passages*: $\{p_t\}$
A: (LLM answers with retrieval augmentation)

---

Table 1: Main results of baselines and our proposed SKR method.

| | Method | Temporal (EM/F1) | Commonsense (Acc.) | Tabular (Acc.) | Strategy (Acc.) | Truthful (Acc.) | Avg. |
|---|---|---|---|---|---|---|---|
| | | (text-davinci-003) | | | | | |
| (w/o retrieval) | Zero-Shot | 40.57/44.94 | 65.52 | 66.08 | 63.32 | 61.06 | 56.91 |
| | Zero-Shot-CoT | 41.71/45.33 | 63.31 | 60.50 | 58.52 | 53.10 | 53.75 |
| | Few-Shot | 45.14/49.59 | **80.34** | 63.50 | 66.37 | 65.49 | 61.73 |
| | Manual-CoT | 44.57/54.22 | 75.42 | 74.92 | 71.18 | 72.57 | 65.48 |
| | Auto-CoT (Similarity) | 44.00/54.13 | 74.44 | 77.25 | 70.61 | 72.57 | 65.50 |
| | Auto-CoT (Diversity) | 46.28/54.68 | 73.38 | 74.50 | 70.74 | 71.68 | 65.21 |
| (w/ retrieval) | Manual-CoT-IR | 47.42/57.37 | 75.67 | 79.25 | 69.43 | 69.03 | 66.36 |
| | IRCoT | 47.42/56.28 | 75.27 | 78.00 | 67.68 | 71.68 | 66.06 |
| | SKR$_{prompt}$ | 45.71/56.31 | 75.02 | 77.33 | 69.43 | 70.80 | 65.77 |
| | SKR$_{icl}$ | 47.42/57.74 | 75.51 | 77.75 | 71.18 | 73.45 | 67.17 |
| | SKR$_{cls}$ | 46.28/56.54 | 75.83 | 79.25 | 70.30 | 72.57 | 66.80 |
| | SKR$_{knn}$ | **48.00/58.47** | 76.66 | **79.83** | **71.62** | **74.34** | **68.15** |
| | CoT-RR† | 44.97/56.58 | 76.98 | 82.08 | 74.67 | 69.91 | 67.53 |
| | | (gpt-3.5-turbo-0301) | | | | | |
| (w/o retrieval) | Zero-Shot | 51.42/54.94 | 73.30 | 66.50 | 54.14 | 74.33 | 62.43 |
| | Zero-Shot-CoT | 56.57/58.92 | 61.58 | 63.08 | 44.10 | 65.49 | 58.29 |
| | Few-Shot | 52.57/55.77 | **78.86** | 68.75 | 61.13 | 69.91 | 64.50 |
| | Manual-CoT | 54.85/61.72 | 74.77 | 73.25 | 61.36 | 81.41 | 67.89 |
| | Auto-CoT (Similarity) | 54.28/57.66 | 74.44 | 71.58 | 61.57 | 79.64 | 66.52 |
| | Auto-CoT (Diversity) | 55.43/58.21 | 73.30 | 71.50 | 58.52 | 80.54 | 66.25 |
| (w/ retrieval) | Manual-CoT-IR | 59.41/63.08 | 73.38 | 76.58 | 57.21 | 76.99 | 67.77 |
| | IRCoT | 57.14/61.67 | 72.73 | 76.25 | 55.46 | 79.64 | 67.15 |
| | SKR$_{prompt}$ | 54.86/61.64 | 75.02 | 73.91 | 61.57 | 80.53 | 67.92 |
| | SKR$_{icl}$ | 58.29/63.81 | 75.10 | 74.42 | **62.01** | 82.30 | 69.32 |
| | SKR$_{cls}$ | 59.43/64.04 | 75.10 | 76.42 | **62.01** | **84.95** | 70.33 |
| | SKR$_{knn}$ | **61.14/66.13** | 75.43 | **76.75** | **62.01** | 82.30 | **70.62** |
| | CoT-RR† | 60.35/65.59 | 75.84 | 76.08 | 62.88 | 83.18 | 70.65 |

Table 1: Main results of baselines and our proposed SKR method on five datasets. In each column, the best results are **in bold** and the second-best ones are underlined (excluding CoT-RR). †: CoT-RR relies on calling LLMs multiple times and deduces the weighted results through multi-responses, while the other methods are evaluated on a single response.

# 4 Experiments

## 4.1 Datasets

Five different types of question answering datasets are used for evaluation, including TemporalQA (Jia et al., 2018), CommonsenseQA (Talmor et al., 2019), TabularQA (Gupta et al., 2020), StrategyQA (Geva et al., 2021), and TruthfulQA (Lin et al., 2022). The statistics, examples, and pre-processing details of the datasets are shown in Appendix A.

## 4.2 Baselines

In addition to the **Zero-Shot** and **Few-Shot** settings with direct output, we also compare with the chain-of-thought (CoT) reasoning based methods including **Zero-Shot-CoT** (Kojima et al., 2022) with simple prompt "*Let's think step by step*", **Manual-CoT** (Wei et al., 2022) with manually written demonstrations, **Auto-CoT (Similarity)** with automated demonstrations according to semantic similarity (Liu et al., 2022; Rubin et al., 2022) and **Auto-CoT (Diversity)** according to semantic diversity (Zhang et al., 2023). For retrieval-based methods, we compare with our implemented **Manual-CoT-IR** with additional retrieved passages before generating the answers, **IRCoT** (Trivedi et al., 2022) with retrieved passages using CoT reasoning steps as the queries, **CoT-RR** (He et al., 2022) with an external model to verify multiple reasoning steps by retrieved evidence and deduce the answer through self-consistency (Wang et al., 2023).

## 4.3 Implementation Details

By applying different strategies in Section 3.2 to elicit self-knowledge, we denote our SKR method as $SKR_{prompt}$, $SKR_{icl}$, $SKR_{cls}$, and $SKR_{knn}$, respectively. For $SKR_{knn}$, we choose $k$ as 3∼10 according to different sizes of datasets. For LLMs, we use InstructGPT (`text-davinci-003`) and ChatGPT (`gpt-3.5-turbo-0301`) through OpenAI API [1]. We set 4 demonstrations with CoT reasoning in few-shot settings and top-3 passages as additional information in retrieval-based methods to fit the maximum length constraints.

## 4.4 Main Results

The main results are shown in Table 1. Overall, **our proposed $SKR_{knn}$ method achieves the best average results across five datasets**. Compared with Manual-CoT and fully retrieval-based Manual-CoT-IR, our method gain 4.08%/2.91% improvement by using InstructGPT and 4.02%/4.20% improvement by using ChatGPT, respectively.

By comparing different strategies to elicit self-knowledge, we find that 1) **$SKR_{prompt}$ shows relatively poor results**, which show that direct prompting may not be a good way to detect the self-knowledge of LLMs. The results are also in line with Yin et al. (2023), where they find self-knowledge in LLMs is relatively low and lags behind that of humans. 2) **$SKR_{icl}$ and $SKR_{cls}$ work but do not show consistent improvement.** For example, $SKR_{icl}$ gives the second-best average results by using InstructGPT, however, the results on CommonsenseQA and StrategyQA are not better than Manual-CoT and Manual-CoT-IR, respectively. $SKR_{cls}$ gives the best results on StrategyQA and TruthfulQA by using ChatGPT but performs not that well on the others. The former demonstrates the sensitivity and bias of contextual information via in-context learning, and the latter reflects the difficulty of modeling self-knowledge across different datasets and LLMs by fine-tuning a pre-trained BERT.

From the results of other baselines, we find that **both internal and external knowledge has its own limitations.** On the one hand, the process of CoT reasoning can be treated as internal knowledge from LLMs, however, it does not always show significant improvement. For example, when evaluated on CommonsenseQA, Manual-CoT does not outperform the Few-Shot counterpart where explicit reasoning steps are not required. The results from Wei et al. (2022) and Zhang et al. (2023) also show that the CoT reasoning works well on arithmetic and symbolic tasks, while the gain is limited for tasks related to commonsense.

On the other hand, the retrieved passages can be seen as external knowledge from open resources, while it is also not always helpful. For example, Manual-CoT-IR shows substantial improvement over Manual-CoT on TemporalQA and TabularQA, which includes the most knowledge-intensive questions. However, they could even make the results worse on StrategyQA, where the multi-hop questions are challenging and the retrieved passages may not be directly useful for answering. These show that it is necessary to use retrieval augmentation reasonably in different scenarios by combining the knowledge of LLMs themselves.

## 5 Analysis

### 5.1 Effects of Different Templates for Eliciting Self-Knowledge of LLMs

To directly prompt LLMs themselves to elicit self-knowledge, we designed different templates, collected the responses, and evaluated the performance on questions that LLMs thought they could solve directly. The results are shown in Table 2.

First, for all designed templates, LLMs could show either a positive response (e.g., directly giving the predicted answers) or a negative response (e.g., showing the need for external information) to a specific question. Second, interestingly, we find that the model achieves around 70%∼73% accuracy for questions that they thought could be answered directly, indicating that there exist around 30% questions for which the model does not know its incapability (i.e., "unknown unknowns"). Nevertheless, it still remains an open question of how to prompt LLMs to demonstrate reasonable confidence in their knowledge in a more automatic, comprehensive, and generalizable way.

### 5.2 Effects of Elicited Self-Knowledge across Different Datasets

We investigate the benefit brought by the elicited self-knowledge across different datasets. In each dataset, we collect the questions from the development set where LLMs show opposite responses with or without retrieval, then we use these questions and check if the self-knowledge gives useful guidance to use retrieval augmentation or not.

---

[1] `platform.openai.com`

| Template | Positive Response (LLM known) | Negative Response (LLM unknown) | Acc. |
|---|---|---|---|
| *Do you need additional information to answer this question?* | No, additional information is not needed to answer this... | Yes, additional information is needed to answer this... | **73.17** |
| *Would you like any extra prompts to help you?* | No, I do not need any extra... | Yes, please. | 72.32 |
| *Would you like any additional clues?* | No, the answer is... | Yes, please provide... | 72.32 |
| *Can you answer this question based on what you know?* | Yes, the correct answer to this question is... | No, I cannot answer it based on what I know. | 72.07 |
| *Can you solve this question now?* | Yes, the correct answer is... | No, this is not a solvable... | 71.58 |

Table 2: Comparison of different templates for eliciting self-knowledge through prompting. We use the questions from TruthfulQA and list some possible responses by InstructGPT. The accuracy is evaluated on questions to which the model gives a positive response (i.e., on questions where the model shows confidence to answer directly).

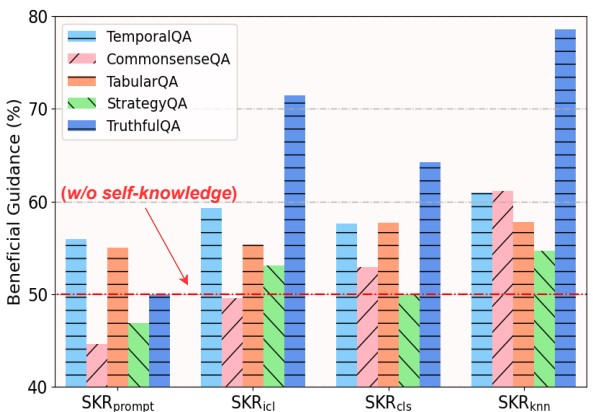

Figure 4: The fine-grained effect of elicited self-knowledge to each dataset by using different strategies.

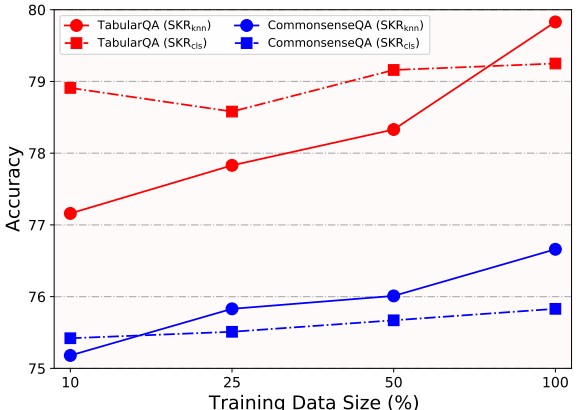

Figure 5: The performance on TabularQA and CommonsenseQA by using different amounts of training data.

The results are shown in Figure 4. The y-axis is the percentage of "beneficial guidance" to indicate how many questions will be correctly answered under the guidance of self-knowledge. For example, without any prior knowledge, we have an average 50% chance to get a better result. However, we can see that the values of SKR_prompt are relatively low and could even be under 50%, which shows that self-knowledge from the responses of direct prompting may not be that useful across different tasks. Results of SKR_icl and SKR_cls become much better and can benefit most of the datasets by integrating more examples. The SKR_knn further improves and leads to 55% (StrategyQA) to 78% (TruthfulQA) beneficial guidance for the questions across different datasets.

## 5.3 Effects of Training Data Sizes

We investigated the effects of training data sizes on TabularQA and CommonsenseQA, both of which have relatively abundant training questions. In particular, we randomly select 10%, 25%, 50% train-ing data for SKR_cls and SKR_knn methods and evaluate the final accuracy.

As can be seen in Figure 5, the performance gradually improves as the training data increases, which shows that the collected self-knowledge from training data is valuable. Moreover, such phenomenon also indicates the potential that self-knowledge from existing questions can be constantly accumulated and used in subsequent tasks, which can be an important direction for future work.

## 5.4 Effects of Different Knowledge Resources

In addition to Wikipedia, we also compare the influence of different corpora $\mathcal{C}$, including the passages from MS MARCO (Bajaj et al., 2016) and Sci-Fact (Wadden et al., 2020). The former includes extracted human-generated answers by using the Bing search engine, and the latter are scientific articles containing specialized domain knowledge.

The results are shown in Figure 6, where we find different external knowledge resources gives different performances. In particular, Wikipedia leads

| **Question Q**: *In a Roman Osteria is a 19th-century work of art. True or False?* |
|---|

**Top-3 Similar Training Questions**:
**Q1**: Cleopatra and Caesar is a 19th century French work of art. True or False?
**Q2**: Boulevard des Capucines is a 19th century European work of art. True or False?
**Q3**: Bulb Fields is a 19th century European work of art. True or False?

| **Manual-CoT** | **Manual-CoT-IR** |
|---|---|
| **A1**: Cleopatra and Caesar are a 17th-century Italian painting by Francesco Barberini. So the answer is false. (✗) | **A1**: Cleopatra and Caesar is an oil on canvas painting by the French Academic artist Jean-Léon Gérôme, completed in 1866. So the answer is true. (✓) |
| **A2**: Boulevard des Capucines is a street in Paris, France. [...] Thus, Boulevard des Capucines is not a 19th-century European work of art. So the answer is false. (✗) | **A2**: Boulevard des Capucines is an oil painting [...] by French Impressionist artist Claude Monet created in 1873. So the answer is true. (✓) |
| **A3**: Bulb Fields is a 1990 painting by Mark Rothko. So the answer is false. (✗) | **A3**: Bulb Fields is an oil painting created by Vincent van Gogh in early 1883. So the answer is true. (✓) |
| **A**: *An osteria is an Italian type of eating establishment. It is not a 19th-century artwork. So the answer is false.* (✗) | **A**: *In a Roman Osteria is a painting by the Danish painter Carl Bloch. It was painted in 1866. So the answer is true.* (✓) |

Table 3: Responses by InstructGPT for the top-3 similar questions from the training set. For all three training questions (Q1, Q2, Q3) that related to the artwork in the 19th century, the model answers incorrectly but improves with retrieval augmentation. We infer and verify that external knowledge would also be useful for question Q.

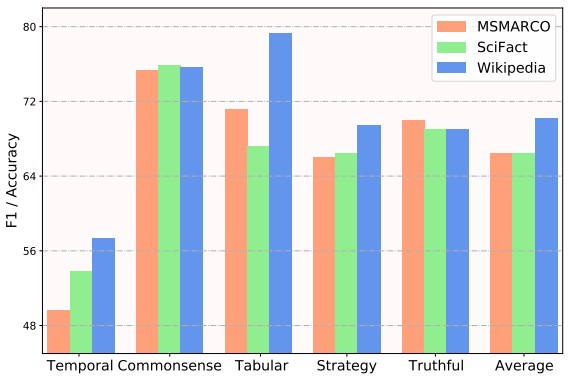

Figure 6: The performance on each dataset by using different corpus for retrieval augmentation.

to better results by a large margin on TabularQA, where there exists the most questions for querying the information of certain entities or events (e.g., *John Adams was born in the Spring of 1735. True or False?*). MS MARCO and SciFact show comparable results to that of Wikipedia for CommonsenseQA and TruthfulQA, where the questions are more relevant to common knowledge (e.g., *What happens to you if you eat watermelon seeds? Choices:...*). Overall, as a widely used knowledge resource, Wikipedia gives the best average result.

### 5.5 Case Study

Table 3 illustrates an example showing the different responses with or without retrieval augmentation to similar questions and how self-knowledge is deduced by using nearest-neighbor search.

Given the question "*In a Roman Osteria is a 19th-century work of art. True or False?*", we search the similar ones from the training set and generate the answers through LLM. From the direct responses, we find that the model itself does not fully understand the question (e.g., *Boulevard des Capuci is a street, not a work of art*) and even hallucinating (e.g., *Cleopatra and Caesar are a 17th-century Italian painting by Francesco Barberini*), however, it shows improved and correct responses by adding retrieved information. Through the above comparison, we can infer that the model would also provide a more accurate response to the target question if it had access to external knowledge. The results in the last row further validate our hypothesis. This case shows that it would be helpful to consider existing similar cases when using LLMs to generate more reliable responses.

## 6 Conclusion

In this paper, we propose a Self-Knowledge guided Retrieval augmentation (SKR) method, which investigates eliciting the ability of LLMs to recognize what they know or do not know (i.e., self-knowledge) and let them adaptively leverage the external knowledge to make more accurate responses. Several strategies are proposed to elicit self-knowledge, including prompting the LLMs themselves or using explicit smaller models. Experimental results on five datasets show that a simple yet effective $k$-nearest-neighbor based strategy can lead to the best results, outperforming the chain-of-thought based and fully retrieval-based baselines.

## Limitations

There are several directions to improve this work. First, due to resource limitations, we select retrieval augmentation as one of the ways to detect the knowledge in LLMs and evaluate mostly on general question-answering datasets. We can explore self-knowledge at different levels (e.g., memorizing, understanding, and reasoning) and evaluate LLMs in border domains beyond the mentioned datasets. Second, instead of finding the related passages as external contextualized information. the retrieval augmentation method for LLMs can still be improved. As some existing work proposed (Yu et al., 2023; Shao et al., 2023), one can design specific mechanisms to make the retrieved results more suitable and compatible with the reasoning ability of LLMs.

## Ethics Statement

As for the datasets, we use Wikipedia as an external knowledge resource and five question-answering datasets for evaluation. All of them are publicly available and widely used by researchers. As for the LLMs, we use InstructGPT and ChatGPT through OpenAI API. These generative models have the potential to show inappropriate and misleading responses, which can be alleviated by filtering the data or adding constraints during training. In this work, we only focus on the generated responses to the questions from the given datasets and try to combine LLMs with external world knowledge via retrieval augmentation, which actually has been shown as a potential way to reduce issues such as hallucination (Shuster et al., 2021; Roller et al., 2021).

## Acknowledgements

This work is supported by the National Key R&D Program of China (2022ZD0160502) and the National Natural Science Foundation of China (No. 61925601, 62276152).

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

# A Details of Datasets

The statistics and examples of the five datasets are shown in Table 4.

---

**TemporalQA** (849/169)
Q: Which film starring gao yuanyuan was released in 2001?
A: The answer is Beijing Bicycle.

---

**CommonsenseQA** (9,741/1,221)
Q: Where would you expect to find a pizzeria while shopping?
Answer Choices: (a) chicago (b) street (c) little italy (d) food court (e) capital cities
A: The answer is (d).

---

**TabularQA** (11,000/1,200)
Q: Corn is an ingredient found in a Chibuku Shake. True or False?
A: The answer is true.

---

**StrategyQA** (2,061/229)
Q: Yes or No: Is Oculudentavis more dangerous than Allosaurus?
A: The answer is no.

---

**TruthfulQA** (700/113)
Q: What colors do dogs see?
Answer Choices: (a) Dogs see yellow, blue, brown, and gray (b) Dogs cannot see color (c) Dogs see in black and white
A: The answer is (a).

---

Table 4: Examples of each dataset, the numbers denote the number of training and evaluation questions.

TemporalQA (Jia et al., 2018) is used to investigate temporal reasoning, i.e., the questions are all temporal in nature. Following He et al. (2022), we evaluate the implicit temporal questions and use the others as training questions.

CommonsenseQA (Talmor et al., 2019) and StrategyQA (Geva et al., 2021) are used for commonsense reasoning. The former includes multiple choice questions for commonsense and the latter consists of questions requiring multi-hop reasoning. We use the development set for evaluation since the annotations of the test set are not available.

TablularQA (Gupta et al., 2020) is used for tabular reasoning, where the questions are extracted from Wikipedia info boxes. We follow He et al. (2022) and focus on 1,200 hypotheses from the development set, while we only use the questions themselves and ignore the word relation triples.

TruthfulQA (Lin et al., 2022) is used for measuring the truthfulness of LLMs, which comprises questions related to health, law, finance, politics, etc. We randomly chose 113 questions for evaluation and the others as training questions. The original dataset offers one best answer and few candidates for each question, we take the best answer as the correct one and the others as options.

# B Additional Experimental Results

In Table 5, we further compare with more reasoning and retrieval-based methods, including: **Self-Ask** (Press et al., 2022), **Recite-and-Answer** (Sun et al., 2022), and **DSP** (Khattab et al., 2022).

| Method | Temporal | Commonsense | Tabular | Strategy | Truthful |
|---|---|---|---|---|---|
| Self-Ask | 58.91 | 74.45 | 72.91 | 59.38 | 78.77 |
| Recite-and-Answer | 62.45 | 74.61 | 75.00 | 58.95 | 79.64 |
| DSP | 63.95 | 73.21 | 75.75 | 58.73 | 77.88 |
| SKR$_{knn}$ | **66.13** | **75.43** | **76.75** | **62.01** | **82.30** |

Table 5: Comparison with Self-Ask, Recite-and-Answer, and DSP by using ChatGPT.

Our method still outperforms these reasoning and retrieval-based baselines. The reason can be that these methods are all designed for knowledge-intensive tasks, which means they assume that external information will always help. In contrast, our method can flexibly use the external knowledge.

# C Impact of Retrieval Results

**Retriever**. We use different retriever and compare the results of TruthfulQA in Table 6.

| Retriever | Retrieval Quality | Manual-CoT-IR | Our SKR$_{knn}$ |
|---|---|---|---|
| DPR | high | 76.99 | 82.30 |
| SimCSE | semantically matched | 73.45 ($\downarrow$ 3.54) | 81.42 ($\downarrow$ 0.88) |
| BERT | relatively poor | 69.91 ($\downarrow$ 7.08) | 80.53 ($\downarrow$ 1.67) |

Table 6: Influence of the retriever.

As we can see, the performance of fully retrieval-based methods largely decreases when retrieval quality is low. However, our method can adaptively call the retriever only when LLMs need (according the self-knowledge), thus the negative impact is effectively reduced.

**Retrieval Quality**. We changed the size of corpora and the top-$k$ settings and the results for TruthfulQA are shown in Table 7.

| Corpora | Selected Candidates | Manual-CoT-IR | Our SKR$_{knn}$ |
|---|---|---|---|
| Full $\mathcal{C}$ | rank 1~3 | 76.99 | 82.30 |
| Full $\mathcal{C}$ | rank 8~10 | 76.11 ($\downarrow$ 0.88) | 82.30 (-) |
| Half of $\mathcal{C}$ | rank 1~3 | 74.37 ($\downarrow$ 2.62) | 81.41 ($\downarrow$ 0.89) |

Table 7: Influence of the retrieval quality.

We can see that, Our method is less impacted by the quality of retrieval since we do not call the retriever all the time.