# OpenReview forum: "Self-Knowledge Guided Retrieval Augmentation for Large Language Models"
_EMNLP/2023/Conference — EMNLP 2023 Findings_

### Official Review · Reviewer_spcX · 2023-07-21

**Soundness:** 3

**Excitement:**

3: Ambivalent: It has merits (e.g., it reports state-of-the-art results, the idea is nice), but there are key weaknesses (e.g., it describes incremental work), and it can significantly benefit from another round of revision. However, I won't object to accepting it if my co-reviewers champion it.

**Paper Topic And Main Contributions:**

This study probes the intriguing question of detecting a language model's knowledge gaps prior to integrating external knowledge. The authors introduce a unique approach that prompts retrieval only when necessary, thereby enhancing the model's efficiency and relevance in knowledge acquisition.

**Questions For The Authors:**

1. Given that we often don't have access to the complete training dataset, or the subset of scenarios related to our questions, and considering the potentially enormous size of these datasets, how do you approach finding the necessary subset to cover known-unknown predictions? How do you manage to achieve this with limited coverage of the dataset?

2. In a more prevalent scenario, models typically don't lack complete knowledge but may lack confidence in certain aspects. How does your approach address this partial uncertainty?

**Reasons To Accept:**

1. The innovative concept of adaptive retrieval, which involves determining what the model already knows before executing a task, presents a novel approach to this field of study.

2. The effectiveness of this method is convincingly demonstrated through empirical evidence, further underscoring its potential as a valuable tool in the advancement of Large Language Models.

**Reasons To Reject:**

1. The approach for determining what the LLM knows seems straightforward, it relies on having access to its training dataset. This could pose implementation challenges in real-world scenarios, where such data may not be readily available.

2. The formulation of the question in the current work seems fairly elementary. In a multitude of real-world scenarios, an LLM might only require additional knowledge about a specific component referred to in the question. In other words, when given a question, an LLM could potentially have partial knowledge while being confused about other aspects. This presents a more nuanced situation, different from the binary scenario of the model knowing everything or nothing at all. (e.g., what is the origin A in B? The model might know about A but not about B, but the proposed method would retrieve the knowledge with the whole query rather than the part the LLM does not know.)

**Reproducibility:**

4: Could mostly reproduce the results, but there may be some variation because of sample variance or minor variations in their interpretation of the protocol or method.

**Reviewer Confidence:**

4: Quite sure. I tried to check the important points carefully. It's unlikely, though conceivable, that I missed something that should affect my ratings.

---

> ### Author Rebuttal · Authors · 2023-08-29
>
> Thanks for your acknowledgment of *"novel approach"*, *"convincingly demonstrated through empirical evidence"*, *"potential as a valuable tool"*. We notice that there might be some misunderstanding regarding our method and we address your concerns below.
>
> > **Q1.** The approach for determining what the LLM knows seems straightforward, it relies on having access to its training dataset. This could pose implementation challenges in real-world scenarios, where such data may not be readily available.
>
> The training dataset $\mathcal{D}$ in our paper is the QA pairs in training set (lines 161-162), not the training corpora of the LLMs.
>
>
> > **Q2.** The formulation of the question in the current work seems fairly elementary. (e.g., what is the origin A in B? The model might know about A but not about B, but the proposed method would retrieve the knowledge with the whole query rather than the part the LLM does not know.)
>
> Thanks for your thoughtful comments and we also agree that the way to retrieve can also be improved (lines 522-524). However, we do not think it should be the reason to reject.
>
> First, taking the whole query as the key for retrieval is widely used across various retrieval-augmented methods, such as REALM[1], RAG[2], FiD[3], etc. Such a way of retrieval is also used in the baselines, so the comparison can be fair.
>
> Second, to our knowledge, the way to retrieve the uncertain part has not been used in existing literature.
>
> Third, the main contribution of this work is to elicit the self-knowledge of LLMs, which can be used to find the part that LLM does not know. Thus our method actually has the potential to improve the way of retrieval and your suggestion can be our future work.
>
> [1] Guu et al. REALM: retrieval-augmented language model pre-training. ICML 2020.
> [2] Lewis et al. Retrieval-Augmented Generation for Knowledge-Intensive NLP Tasks. NeurIPS 2020.
> [3] Izacard et al. Leveraging Passage Retrieval with Generative Models for Open Domain Question Answering. EACL 2021.
>
> > **Q3.** Given that we often don't have access to the complete training dataset, or the subset of scenarios related to our questions, and considering the potentially enormous size of these datasets, how do you approach finding the necessary subset to cover known-unknown predictions? How do you manage to achieve this with limited coverage of the dataset?
>
> First, we think that the lack of enough training dataset is a more general question, which is beyond the scope of this work. For example, if the training set of the existing QA system is limited and can not cover the questions in real scenarios, the final performance will also drop. Therefore, this is a common issue.
>
> Second, similar to most supervised NLP tasks, the performance of our approach increases with more training data (Section 5.3). Thus, beyond the limited training data, we can create the necessary subset to cover known-unknown predictions in a data-augmentation manner. For example, 1) exploring relevant questions in cross-task datasets. 2) generating related questions as external training samples by
> leveraging the LLMs' generation ability.
>
> > **Q4.** In a more prevalent scenario, models typically don't lack complete knowledge but may lack confidence in certain aspects. How does your approach address this partial uncertainty?
>
> Although solving the "partial uncertainty" is not the main target of this paper, we think our approach can still apply to such a scenario.
>
> In particular, instead of asking the model for the binary responses, we can design prompts (e.g., (i) *Please list any additional information you need to answer this question*. (ii) *Which part do you need additional information to answer the question*? (iii) *Which aspect of the question requires further information for a complete response*?) for detailed results. For example, we list some responses by ChatGPT below:
>
> |  |Please list any additional information you need to answer this question. Q: Did any country in Portuguese Colonial War share Switzerlands role in WWII?|
> |  :----  | :----  |
> | Response 1  | I need more information about what specifically you mean by **"Switzerland's role in WWII."** Do you mean the fact that Switzerland was neutral during the war, or do you mean something else?|
> | Response 2  | I need additional information on what specifically you mean by **"Switzerland's role in WWII"** in order to answer this question accurately.|
> | Response 3  | As an AI language model, I do not have personal opinions or beliefs. However, I need more information to answer your question accurately. Please clarify what you mean by **"Switzerland's role in WWII."**|
>
> Such results can be used as specific queries for retrieval and collected for fine-grained self-knowledge acquisition.

---

### Official Review · Reviewer_w5Ym · 2023-08-02

**Typos Grammar Style And Presentation Improvements:** N/A
**Soundness:** 3

**Excitement:**

3: Ambivalent: It has merits (e.g., it reports state-of-the-art results, the idea is nice), but there are key weaknesses (e.g., it describes incremental work), and it can significantly benefit from another round of revision. However, I won't object to accepting it if my co-reviewers champion it.

**Missing References:**

N/A

**Paper Topic And Main Contributions:**

This paper investigates eliciting the self knowledge of LLMs and propose Self-Knowledge guided Retrieval augmentation (SKR) method to flexibly call the retriever for making better use of both internal and external knowledge. Experimental results show that SKR outperforms chain-of-thought based and fully retrieval-based methods.

**Questions For The Authors:**

See Reasons To Reject

**Reasons To Accept:**

(1) Propose a novel method to elicit the self-knowledge of LLMs to identify what LLMs know and don't know.
(2) This paper is well written and easy to follow.

**Reasons To Reject:**

(1) More knowledge-intensive datasets should be considered, such as the datasets in KILT.
(2) Missing IR-augmented baselines such as Self-Ask and DSP.
(3) Missing reasoning baselines such as Recite-and-answer.
(4) Unfair comparison, the proposed method needs additional training data but baselines do not.

**Reproducibility:**

4: Could mostly reproduce the results, but there may be some variation because of sample variance or minor variations in their interpretation of the protocol or method.

**Reviewer Confidence:**

3: Pretty sure, but there's a chance I missed something. Although I have a good feel for this area in general, I did not carefully check the paper's details, e.g., the math, experimental design, or novelty.

---

> ### Author Rebuttal · Authors · 2023-08-29
>
> Thanks for your valuable feedback. We see that your main concerns are baseline settings. We now address them as follows.
>
> > **Q1.**  More knowledge-intensive datasets should be considered, such as the datasets in KILT.
> **Q2.**  Missing IR-augmented baselines such as Self-Ask and DSP.
> **Q3.** Missing reasoning baselines such as Recite-and-answer.
>
> Below we report the additional experimental results (by gpt-3.5-turbo-0301) according to your suggestions.
>
>
> |      | Temporal | Commonsense  | Tabular  |  Strategy  | Truthful | FEVER  |  Avg. |
> |  :----  | :----:  |  :----:  | :----:  |  :----:  | :----:  |  :----:  | :----:  |
> |Manual-CoT   |	61.72|	74.77|	73.25|	61.36|	81.41  |56.60|68.18|
> |Manual-CoT-IR   |	63.08|	73.38|	76.58|	57.21|	76.99 |60.20|67.91|
> | Self-Ask  |58.91|	74.45|	72.91|	59.38|	78.77 |56.20|66.77|
> | Recite-and-answer  |	62.45|	74.61|	75.00|	58.95|	79.64 |60.40|68.51|
> | SKR$_{knn}$  |	**66.13**|	**75.43**|	**76.75**|	**62.01**|	**82.30** |**61.40**|**70.67**|
>
> First, we add the FEVER (fact verification) dataset in KILT for evaluation. We sampled 5,000/500 claims as train/dev sets and our method gives the best accuracy (61.40) compared with baselines, which shows that our method can still apply to knowledge-intensive tasks. We'll consider adding more tasks from KILT benchmark in the future.
>
> Second, results show that our method still outperforms baselines such as Self-Ask and Recite-and-answer across different tasks.
>
> Third, we find that your mentioned work (Self-Ask, DSP, and Recite-and-answer) are all designed for knowledge-intensive tasks, where they rely on high-quality external knowledge for all time. Our method complementarily pays more attention to the internal knowledge of LLMs and can be applied to more general QA scenarios.
>
> In all, we thank you for pointing out these relevant works. We are willing to incorporate the aforementioned results and discussions into our work.
>
>
> > **Q4.**  Unfair comparison, the proposed method needs additional training data but baselines do not.
>
> Not exactly. 1) For the training QA datasets, all baselines (except zero-shot setting) also select examples from training data for in-context-learning. Further, Auto-CoT uses the full training data to build the demonstrations for each test instance. 2) For the retrieval data, all the retrieval-augmented baselines and our method use the same corpora. Therefore, the comparison should be fair.

---

### Official Review · Reviewer_KHre · 2023-08-03

**Soundness:** 3

**Excitement:**

4: Strong: This paper deepens the understanding of some phenomenon or lowers the barriers to an existing research direction.

**Paper Topic And Main Contributions:**

This paper describes SKR (Self-Knowledge guided Retrieval augmentation), which is a collection of strategies for deciding whether to retrieve given the model's "self-knowledge" (i.e. whether it believes it "knows" the answer). The paper describes 4 strategies: 1). SKR_prompt: directly asking the model whether it knows the answer, 2). SKR_icl: providing examples of questions that the model knows or does not know the answer to in the prompt, 3). SKR_cls: training a small model to predict whether the answer to a given question is known or not known by the model (weights and bias on top of a BERT base sentence embedding of the question trained against a cross-entropy loss), and 4). SKR_knn: classification of whether the model knows the answer based on the model's knowledge of the answers to the k most similar questions according to a KNN algorithm. This collection of approaches (and SKR_knn in particular) yields modest but consistent improvements over retrieval augmented and non-retrieval augmented baselines, including Zero-Shot and Few-Shot inference, CoT approaches such as Zero-Shot-CoT, Manual-CoT, and Auto-CoT (both Similarity and Diversity), Manual-CoT-IR, and IRCoT. In some cases, CoT-RR outperforms SKR, but it also requires multiple calls to the LLM and multiple responses, rendering this an unfair comparison.

**Reasons To Accept:**

+ This paper explores (and does a good job of motivating) an interesting and important problem, namely the extent to which models are aware of their own knowledge and how we might leverage this self-knowledge if present.
+ It reports decent improvements across five different question-answer datasets (~4% improvement over Manual-CoT methods with/without retrieval).
+ Interesting analyses, e.g. (~70% accuracy on questions the model claims to know the answer to (Section 5.1), exploration of the effect of training data sizes and knowledge sources (5.3 and 5.4).


**Reasons To Reject:**

- SKR would be more useful if there were a single self-knowledge strategy that was consistently effective across datasets.
- The paper would have benefited from more exploration of the impact of the retrieval quality, e.g. as the number of retrieved passages increases, does the response accuracy decline due to lower retrieval quality? Is there a different "sweet spot" depending on model confidence?
- The case study was not convincing in this reviewer's opinion, and probably belonged in the appendix rather than the main paper. As a human reader, this question is quite confusing. The results would likely have been quite different if "In a Roman Osteria" had been enclosed in quotes or some other delimiter.

**Reproducibility:**

3: Could reproduce the results with some difficulty. The settings of parameters are underspecified or subjectively determined; the training/evaluation data are not widely available.

**Reviewer Confidence:**

4: Quite sure. I tried to check the important points carefully. It's unlikely, though conceivable, that I missed something that should affect my ratings.

**Typos Grammar Style And Presentation Improvements:**

- The paper did a good job of motivating the overall problem, but was not carefully written in some places. Additional motivation / clarity would be beneficial in a number of sections (e.g. it is not clear what is meant by "Direct prompting may intuitively work, but it tests each question independently and does not make use of the collected training questions in Section 3.1" or "Second, in-context learning could also be unstable due to contextual bias and sensitivity."). As another example, Section 4.1 provides no justification for the choice of Q&A datasets.
-"The retrieved passages are relevant but not much helpful" -> "not particularly helpful"
-"can not" -> "cannot" (multiple places)
-"We use the LLM itself in the former two methods and explicit smaller modes in the latter two methods" -> "models"
-"LLM direct answers without retrieval" -> "directly" (multiple places)
-"SKR_icl and SKR_cls work but does not show consistent improvement" -> "do not"
-"In addition to Wikipedia, we also compare the influence of different corpus C" -> "corpora"

---

> ### Author Rebuttal · Authors · 2023-08-29
>
> Thanks for your acknowledgment of *"good job of motivating"*, *"interesting and important problem"*, *"decent improvements"*, *"interesting analyses"*. We now address your concerns below.
>
> > **Q1.** SKR would be more useful if there were a single self-knowledge strategy that was consistently effective across datasets.
>
> We agree. Actually our method SKR$_{knn}$ achieves the best results across almost all datasets. The only exception is CommonsenseQA, where the performance of Few-Shot is extremely high. Such a result is also reported in refs[1][2], and we also explained this in lines 373-380.
>
> [1] Wei et al. Chain-of-thought prompting elicits reasoning in large language models. NeurIPS 2022.
> [2] Zhang et al.  Automatic chain of thought prompting in large language models. ICLR 2023.
>
>
> > **Q2.** The paper would have benefited from more exploration of the impact of the retrieval quality, e.g. as the number of retrieved passages increases, does the response accuracy decline due to lower retrieval quality? Is there a different "sweet spot" depending on model confidence?
>
> Thanks for your constructive comments. We changed the size of corpora and the top-K settings according to your suggestion, the results for TruthfulQA are shown below.
>
> | Corpora | Selected Passages | Manual-CoT-IR |Our SKR$_{knn}$ |
> |  :----:|  :----: | :----:  | :----: |
> |full $\mathcal{C}$|rank 1-3 candidates |76.99|82.30|
> |full $\mathcal{C}$|rank 8-10 candidates |76.11($\downarrow$ 0.88) | 82.30 (-)|
> |half of $\mathcal{C}$|rank 1-3 candidates|74.37 ($\downarrow$ 2.62) |81.41 ($\downarrow$ 0.89)|
>
> We can see that, compared with the fully retrieval-based methods, our method is less impacted by the quality of retrieval since we do not call the retriever all the time. In extreme cases, if the retrieved results are harmful (e.g., useless noise), the elicited self-knowledge will guide the model to answers directly according to its parametric knowledge, and the accuracy will not decrease rapidly.
>
> > **Q3.** The case study was not convincing in this reviewer's opinion, and probably belonged in the appendix rather than the main paper.
>
> We will consider your suggestion and strive to incorporate more valuable and easily understandable results into the main pages.
>
> > **Q4.** As a human reader, this question is quite confusing. The results would likely have been quite different if "In a Roman Osteria" had been enclosed in quotes or some other delimiter.
>
> That's a good point and we try adding some quotes and other delimiter to check the difference. However, we find that the model still regularly generates hallucinations and answers incorrectly. For example,
>
> |  |"In a Roman Osteria" is a 19th-century work of art. True or False? |
> |  :----  | :----  |
> | Response 1 | "In a Roman Osteria" is a painting created by Eduardo Dalbono in 2009. So the answer is false. |
> | Response 2 | "In a Roman Osteria" is an oil on canvas by the Dutch painter Jan Steen. The date of work is 1660. So the answer is false. |
> | Response 3 | "In a Roman Osteria" is a painting by Italian artist Pietro Annigoni that was created in 1954. So the answer is false. |
> ||**\<In a Roman Osteria\> is a 19th-century work of art. True or False?**|
> | Response 1 | \<In a Roman Osteria\> is an oil on canvas painting by Dutch painter Jan Steen, was completed in 1668. So the answer is false. |
> | Response 2 | \<In a Roman Osteria\> is a painting by the Dutch artist Pieter Bruegel the Elder, created in 1569. So the answer is false. |
> | Response 3 | \<In a Roman Osteria\> is an 18th-century painting created by Giuseppe Maria Crespi. So the answer is false. |
>
> > **Q5.** The paper did a good job of motivating the overall problem, but was not carefully written in some places.
>
> Thank you for pointing out these typos. This will definitely help us improve our work.

---

### Official Review · Reviewer_jnMH · 2023-08-04

**Soundness:** 3

**Excitement:**

4: Strong: This paper deepens the understanding of some phenomenon or lowers the barriers to an existing research direction.

**Paper Topic And Main Contributions:**

This paper introduces a new method of SKR to flexibly call the retriever for making better use of both internal and external knowledge. Specifically, a memory bank for positive/negative samples is maintained. And the new method detects LLMs’ self-knowledge by training a classifier or using context learning based on the memory bank. At last, the method is tested on five different types of question answering datasets and proven to achieve noticeable improvements.


**Questions For The Authors:**

1. It would be interesting to see SKR’s performance using different retrievers since the selection of positive/negative and the retrieval augmentation both depend on them. I have a hunch that this method relies heavily on how strong the retriever is.
2. Is Figure 4 just the same result in Table 1 with a different illustration? How is the performance of the self-knowledge elicitation of LLMs in terms of the four proposed strategies? Does their performance affect the final performance?
3. The experimental datasets do not involve any reasoning tasks, which raises the question of why CoT was employed. Moreover, it seems the zero-shot CoT generally exhibits performance degradation compared to the zero-shot baseline.

**Reasons To Accept:**

1. The paper is well-organized and is easy to understand.
2. The paper proposes a novel framework for retrieval-augemented LLMs.
3. The paper evaluates different variants of the method and most of them achieve better results than baselines. The experiments in the paper are extensive and convincing. The chosen evaluation metrics are presentative, and ablation experiments show the efficacy of the SKR's design choices.


**Reasons To Reject:**

1. The advantages of SKR_knn over the baseline method (Manual-CoT-IR) do not appear to be significant, with an improvement of less than one point observed in 3 out of 5 datasets (Temporal, Commonsense, and Tabular). What could be the underlying reasons for this limited enhancement?
2. The author's analysis lacks an examination of the effectiveness of various self-knowledge methods. Relying solely on the final QA performance does not offer good support for this point. A more persuasive approach would involve comparing the accuracy, precision, recall, and other relevant metrics of these different methods. Unfortunately, the study did not include such experiments.


**Reproducibility:**

3: Could reproduce the results with some difficulty. The settings of parameters are underspecified or subjectively determined; the training/evaluation data are not widely available.

**Reviewer Confidence:**

4: Quite sure. I tried to check the important points carefully. It's unlikely, though conceivable, that I missed something that should affect my ratings.

---

> ### Author Rebuttal · Authors · 2023-08-29
>
> Thanks for your kind words such as *"well-organized paper"*, *"novel framework"*, *"extensive and convincing experiments"*. We see that your main concerns are some experimental settings. We now answer the questions below.
> > **Q1.** The advantages of SKR_knn over the baseline method (Manual-CoT-IR) do not appear to be significant, with an improvement of less than one point observed in 3 out of 5 datasets (Temporal, Commonsense, and Tabular).
>
> First, the performance advantages over the Manual-CoT-IR baseline are 1.10/0.99/0.58 (for text-davinci-003) and 3.05/2.05/0.17 (for gpt-3.5-turbo-0301) in Temporal, Commonsense, and Tabular datasets, respectively. The overall improvement is not marginal.
>
> Second, besides the numerical advantage in performance, our method can better leverage the internal knowledge of LLMs and reduce the cost of retrieving. For example, by eliciting self-knowledge via a small model, our method can reduce reliance on the retriever for 21%/52%/29% of questions in Temporal, Commonsense, and Tabular datasets, respectively. In practical applications, the time-saving benefits will be further amplified.
>
>
>
> > **Q2.** The author's analysis lacks an examination of the effectiveness of various self-knowledge methods. Relying solely on the final QA performance does not offer good support for this point. A more persuasive approach would involve comparing the accuracy, precision, recall, and other relevant metrics of these different methods.
>
> Besides the final QA performance in Table 1, we defined a metric "**beneficial guidance**" (lines 425-430) and show the results in Figure 4. In particular, it is defined as the proportion of correct choices that the LLMs select whether to retrieve or not. Such "beneficial guidance" can be used to measure how correct the self-knowledge is and understand the effectiveness of different strategies for eliciting self-knowledge. Through experimental results, we can find that the better QA results are largely due to higher "beneficial guidance" (i.e., the right choice). See also the response to Q4.
>
>
> > **Q3.** I have a hunch that this method relies heavily on how strong the retriever is.
>
> We use different strategies to retrieve passages and compare the results of TruthfulQA below.
>
> | Retriever | Retrieval Quality | Manual-CoT-IR |Our SKR$_{knn}$ |
> |  :----  | :----  | :----:  | :----: |
> | DPR[1] | high|76.99|82.30|
> | SimCSE[2] | semantically matched|73.45 ($\downarrow$ 3.54)| 81.42 ($\downarrow$ 0.88)|
> | BERT CLS Token$[3]| relatively poor|69.91 ($\downarrow$ 7.08) |80.53 ($\downarrow$ 1.67)|
>
>
> As we can see, the performance of fully retrieval-based methods largely decreases when retrieval quality is low. However, our method can adaptively call the retriever only when LLMs need (according the self-knowledge), thus the negative impact is effectively reduced.
>
> [1] Karpukhin et al. Dense Passage Retrieval for Open-Domain Question Answering. EMNLP 2020.
> [2] Gao et al. SimCSE: Simple Contrastive Learning of Sentence Embeddings. EMNLP 2021.
> [3] Devlin et al. BERT: Pre-training of Deep Bidirectional Transformers for Language Understanding. NAACL 2019.
>
>
> > **Q4.** Is Figure 4 just the same result in Table 1 with a different illustration? How is the performance of the self-knowledge elicitation of LLMs in terms of the four proposed strategies? Does their performance affect the final performance?
>
> Not exactly the same, the evaluated questions in Figure 4 are those LLMs show opposite responses with or without retrieval (lines 418-424). Questions that LLMs show the same responses with or without retrieval are not included for better analysis.
>
> For the random baseline w/o self-knowledge, it chooses to retrieve or not with 50%/50% probability. However, our methods with different strategies can generally give better or even 70% "beneficial guidance" (i.e., the right choice), which can lead to better QA results in Table 1.
>
>
> > **Q5.** The experimental datasets do not involve any reasoning tasks, which raises the question of why CoT was employed. Moreover, it seems the zero-shot CoT generally exhibits performance degradation compared to the zero-shot baseline.
>
> First, CommonsenseQA[1] and StrategyQA[2] are actually two tasks that require reasoning ability. These two tasks are also evaluated among literature such as CoT[3] and Auto-CoT[4].
>
> Second, to our knowledge, CoT and zero-shot CoT[5] work best for arithmetic tasks. However, these tasks are not suitable for comparison within this study because they do not rely on information from external corpora.
>
> Third, the low performance of zero-shot CoT (only with the prompt "Let's think step-by-step") does not mean CoT is useless. The CoT (or more generally, the rationale augmented reasoning) can still play a role across the tasks. Manual-CoT baselines are still important and worth comparison.
>
> [1] Talmor et al. CommonsenseQA: A Question Answering Challenge Targeting Commonsense Knowledge. NAACL 2019.
> [2] Geva et al. Did Aristotle Use a Laptop? A Question Answering Benchmark with Implicit Reasoning Strategies. TACL 2021.
> [3] Wei et al. Chain-of-thought prompting elicits reasoning in large language models. NeurIPS 2022.
> [4] Zhang et al.  Automatic chain of thought prompting in large language models. ICLR 2023.
> [5] Kojima et al. Large language models are zero-shot reasoners. NeurIPS 2022.

---

### Meta-Review · Area_Chair_4L1L · 2023-09-19

**Recommendation:** 3

**Metareview:**

This paper proposes  to flexibly call the retriever for making better use of both internal and external knowledge. Specifically,  a memory bank for positive/negative samples is maintained. and the new method detects LLMs’ self-knowledge by either training a classifier (SKR_cls) or using context learning based on the memory bank (SKR_knn) or directly asking the model whether it knows the answer (SKR_prompt).

On five different types of question answering datasets the KNN approach achieves improvements over other baselines.

Strength
1. A novel framework for an important problem -- considering when to make retrieval for retrieval-augemented LLMs .
2. The knn approach achieves mostly consistent improvements over retrieval augmented and non-retrieval augmented baselines

Weakness: more comprehensive experiment
1. Not experimented with knowledge-intensive tasks such as KILT.
2. Not compared to baselines in IR-augmented  (Self-Ask and DSP) and reasoning (Recite-and-answer).
3. only one type of retriever is experimented with.
4. Would be helpful to compare to related work that leverages the confidence of retrieval model (SearChain)

Weakness: limitations of the model
1. It relies on access to the training dataset, which may not be readily available.
2. The proposed method would retrieve the knowledge with the whole query rather than the part the LLM does not know.

---

### Decision · Program_Chairs · 2023-10-07

**Decision:**

Accept-Findings

**Comment:**

This paper proposes  to flexibly call the retriever for making better use of both internal and external knowledge. Specifically,  a memory bank for positive/negative samples is maintained. and the new method detects LLMs’ self-knowledge by either training a classifier (SKR_cls) or using context learning based on the memory bank (SKR_knn) or directly asking the model whether it knows the answer (SKR_prompt).

On five different types of question answering datasets the KNN approach achieves improvements over other baselines.

Strength
1. A novel framework for an important problem -- considering when to make retrieval for retrieval-augemented LLMs .
2. The knn approach achieves mostly consistent improvements over retrieval augmented and non-retrieval augmented baselines

Weakness: more comprehensive experiment
1. Not experimented with knowledge-intensive tasks such as KILT.
2. Not compared to baselines in IR-augmented  (Self-Ask and DSP) and reasoning (Recite-and-answer).
3. only one type of retriever is experimented with.
4. Would be helpful to compare to related work that leverages the confidence of retrieval model (SearChain)

Weakness: limitations of the model
1. It relies on access to the training dataset, which may not be readily available.
2. The proposed method would retrieve the knowledge with the whole query rather than the part the LLM does not know.